

# Ensuring reliability in electronic examinations through UPPAAL-based trustworthy design

Wenbo Zhou[1,2,3], Yujiao Zhao[1], Ye Zhang[1], Liwen Mu[1,4], Yiyuan Wang[1,3] and Minghao Yin[1,3]

[1] School of Information Science and Technology, Northeast Normal University, Changchun, China
[2] Guangxi Key Lab of Multi-Source Information Mining & Security, Guangxi Normal University, Guilin, China
[3] Key Laboratory of Applied Statistics of Ministry of Education, Northeast Normal University, Changchun, China
[4] School of Physics, Northeast Normal University, Changchun, China

## ABSTRACT

Electronic examination serves as an efficient method for assessing learning outcomes, yet the integration of computers into exam processes introduces potential for unreliability. In this article, we propose a formal model for electronic examinations using timed automata, providing a structured approach to understanding and managing the complexities. The electronic examination process is modeled by defining four UPPAAL templates, *i.e.*, candidate, administrator, invigilator, and examiner. Crucial properties specific to electronic examination are encoded as specifications in UPPAAL. Verification against these properties demonstrates the validity and reliability of this model. The modelable and verifiable electronic examination designed with UPPAAL suggests great potential for deeper exploration in trustworthy digital education.

## INTRODUCTION

Examinations play a preeminent pedagogical role, enabling individuals to assess their skills and knowledge in a specific subject (*Giustolisi, 2018*). In contemporary education, e-learning has emerged as a prevalent option for universities, facilitating the convenient expansion of teaching or learning activities at any time and from anywhere. Particularly with the outbreak of the epidemic, e-learning has garnered increased attention in the field of education. As a vital component of e-learning, electronic examinations serve an irreplaceable function in facilitating efficient evaluation of learning outcomes.

Electronic examinations offer a convenient means to assess learning outcomes utilizing ubiquitous internet-connected devices. The adoption of online learning management systems has enabled universities to boost enrollments without the need for additional lecture halls (*Ngqondi, Maoneke & Mauwa, 2021*). However, electronic examinations have been met with skepticism, primarily due to concerns about academic fraud (*Ngqondi, Maoneke & Mauwa, 2021*). Introducing computers into various phases of examinations

Corresponding authors
Yiyuan Wang,
wangyy912@nenu.edu.cn
Minghao Yin, ymh@nenu.edu.cn

brings new security challenges (*Bella et al., 2017*). Simultaneously, many learners express apprehensions about potential technical glitches during the implementation of electronic examinations (*Ilgaz & Afacan Adanır, 2020*). In such a scenario, ensuring the reliable, fair, and seamless execution of electronic examinations in e-learning becomes paramount (*Muzaffar et al., 2021*). The design of electronic examination systems should prioritize reliability, taking into account its significance and impact. Current electronic examination systems face inherent limitations, primarily due to a lack of focus on robust system model design. This often results in system designs that struggle to ensure dependable and effective implementation, leading to potential inconsistencies and vulnerabilities. These issues highlight the need for improved system designs that promote reliability in electronic examinations.

Reliability in the context of electronic examinations is pivotal to ensure the seamless execution of exams and uphold fairness. In the realm of education, examination rules are explicitly outlined and must be strictly followed, often enforced in an artificial manner. For electronic examinations, it becomes imperative for such rules to seamlessly integrate with examination systems. Correct-by-construction is among the most effective approaches to guarantee the reliability of software systems right from the source. Formal methods provide robust assurances during system design, ensuring the strict adherence to examination rules. These methods typically involve constructing suitable formal models and verifying them before proceeding to further implementation. Adopting this methodology offers numerous advantages. On one hand, it allows for the identification and correction of errors at an early stage, preventing them from evolving into intricate and challenging faults that are both costly and time-consuming to diagnose and repair. On the other hand, formal methods employ a range of proving or model-checking techniques. Notably, executable models can be directly simulated and analyzed, enabling the exploration of diverse design options and verification possibilities (*Bobba et al., 2018*).

To construct an abstract and executable model, the chosen formal method must be appropriate and highly applicable. As one of the most widely employed formal methods, timed automata are a valuable formalism for verifying concurrent systems under timing constraints (*Arcile & André, 2022*). It has been extensively applied to modeling and analysis of many critical systems as well as assistance in the generation of new methods (*Hofmann & Schupp, 2023*; *Lehmann & Schupp, 2022*), including various industrial applications (*Basile et al., 2021*; *Sakata et al., 2023*). One of the most significant advantages of time automata lies in their comprehensibility through graphical representation. Another strength is their executability with timing constraints, enabling the exploration of diverse scenarios and the validation of critical properties. Furthermore, well-established support tools such as UPPAAL (*Behrmann, David & Larsen, 2004*), SPIN (*Holzmann, 2004*), NuSMV (*Cimatti et al., 2002*) and PAT (*Sun et al., 2009*), contribute substantial capabilities to practical modeling, simulation, and verification. UPPAAL is developed in collaboration between the Department of Information Technology at Uppsala University (UPP) in Sweden and the Department of Computer Science at Aalborg University (AAL) in Denmark, with input from several other universities around the world (*Uppsala University & Aalborg University, 2021*). Due to its wide application and powerful

modeling capabilities, we use UPPAAL to model and analyze the electronic examination system.

In this article, we introduce a model for an electronic examination system based on timed automata. We conduct simulations and analyses of crucial properties using the UPPAAL tool. The UPPAAL model we developed is archived and can be found at our GitHub repository (*Zhou, 2024*). To the best of our knowledge, this is the initial attempt to model examinations using UPPAAL. Our UPPAAL-based framework offers a structured method to understand and manage the complexities inherent in electronic examinations. By leveraging the modeling and verifiable capabilities of UPPAAL, there exists a potential for assessing and enhancing electronic examinations in a precise manner. This electronic examination model can function as a foundational component, offering more possibility for further exploration in the realm of trustworthy digital education. To save space, we assume the reader has a fundamental understanding of the UPPAAL tool. More detailed definitions regarding the syntax and semantics of languages used in UPPAAL can be referenced in *Behrmann, David & Larsen (2004)*. We note that cryptography aspects are not within the scope of this article. The novel contributions of this study include:

- Developing a formal model for electronic examinations based on timed automata.
- Encoding typical properties related to electronic examinations as specifications in UPPAAL.
- Verifying our model against these properties using UPPAAL to demonstrate the reliability of electronic examinations.

The significance and impact of this research are multifaceted. By developing a formal model for electronic examinations, our approach tries to enhance the effectiveness of digital assessments to some extent, which is particularly relevant with the increased reliance on e-learning. We acknowledge that our approach is preliminary and foundational, but it is a potential attempt to apply UPPAAL in the education field. Our model can serve as a basic framework for further research in statistical model checking and temporal analysis, aiding in the practical development of more reliable electronic examination systems.

The remainder of this article is structured as follows. "Preliminaries" provides a brief overview of the background on electronic examinations and UPPAAL. "Modeling electronic examination using UPPAAL" details the model specification and property specification for electronic examination. In "Validation and verification in UPPAAL", we analyze and check typical properties using UPPAAL. "Related work" presents a comparison of related research. "Discussions" covers the experience with UPPAAL, limitations and potential biases, and ethical and privacy aspects. Finally, "Conclusions" summarizes this article and outlines future work.

## PRELIMINARIES

In this section, fundamental concepts about electronic examinations and UPPAAL are introduced to enhance comprehension of the following models. Regarding electronic

examinations, we illustrate the roles of participants and their interactions. As for UPPAAL, a concise description of its core components is provided.

## Electronic examination

Electronic examinations offer a convenient way for assessing the knowledge and abilities of learners with the help of computer and network technologies. The participants in electronic examinations are similar to those in traditional examinations, with the difference that certain operations are conducted through network communications.

### Roles of participants

In a general way, there are four roles in an electronic examination, *i.e.*, candidate, administrator, invigilator, and examiner. We list their functions as follows.

- *Candidate*: A candidate is a student taking the examination.
- *Administrator*: An administrator is responsible for registering candidates for the examination.
- *Invigilator*: An invigilator is tasked with distributing questions, supervising the examination and collects answers.
- *Examiner*: An examiner marks the examination and notifies students their scores.

Each participant is assigned specific tasks, and through collaborative efforts, they contribute to the successful execution of electronic examinations.

### Interactions among participants

This subsection explores the dynamic interactions that take place among the various participants involved in the electronic examination process. As depicted in Fig. 1, we use the UML sequence chart notation to illustrate the interactions among participants in the electronic examination process. Prior to the commencement of an electronic examination, the examiner must establish correct answers for the questions (`corrAns`). The sequence unfolds as follows: First, a candidate, denoted as $c_i$, registers with the administrator (`register`). Then, $c_i$ logs into the examination system and notifies the invigilator (`login`). Next, the invigilator dispatches a question to $c_i$ (`get`), who, in turn, formulates and submits an answer (`submit`). Upon receiving the answer, the invigilator confirms the submission through acknowledgment (`accept`). This iterative process may repeat multiple times according to the examination time and the number of questions. When the candidate's examination time expires or the candidate has completed all their answers, they await their scores. Afterward, the invigilator notifies the examiner to commence grading (`mark`) and inform $c_i$ of his/her scores (`notify`). Finally, the candidates, invigilator and examiner complete their processes (`end`), and the administrator resets the system (`reset`). The entire examination process ends when all candidates have been notified of their scores. This ensures that the system correctly captures and evaluates all responses before terminating the examination. The reset ensures that all candidates, the invigilator, and the examiner are reset simultaneously. This is achieved by the

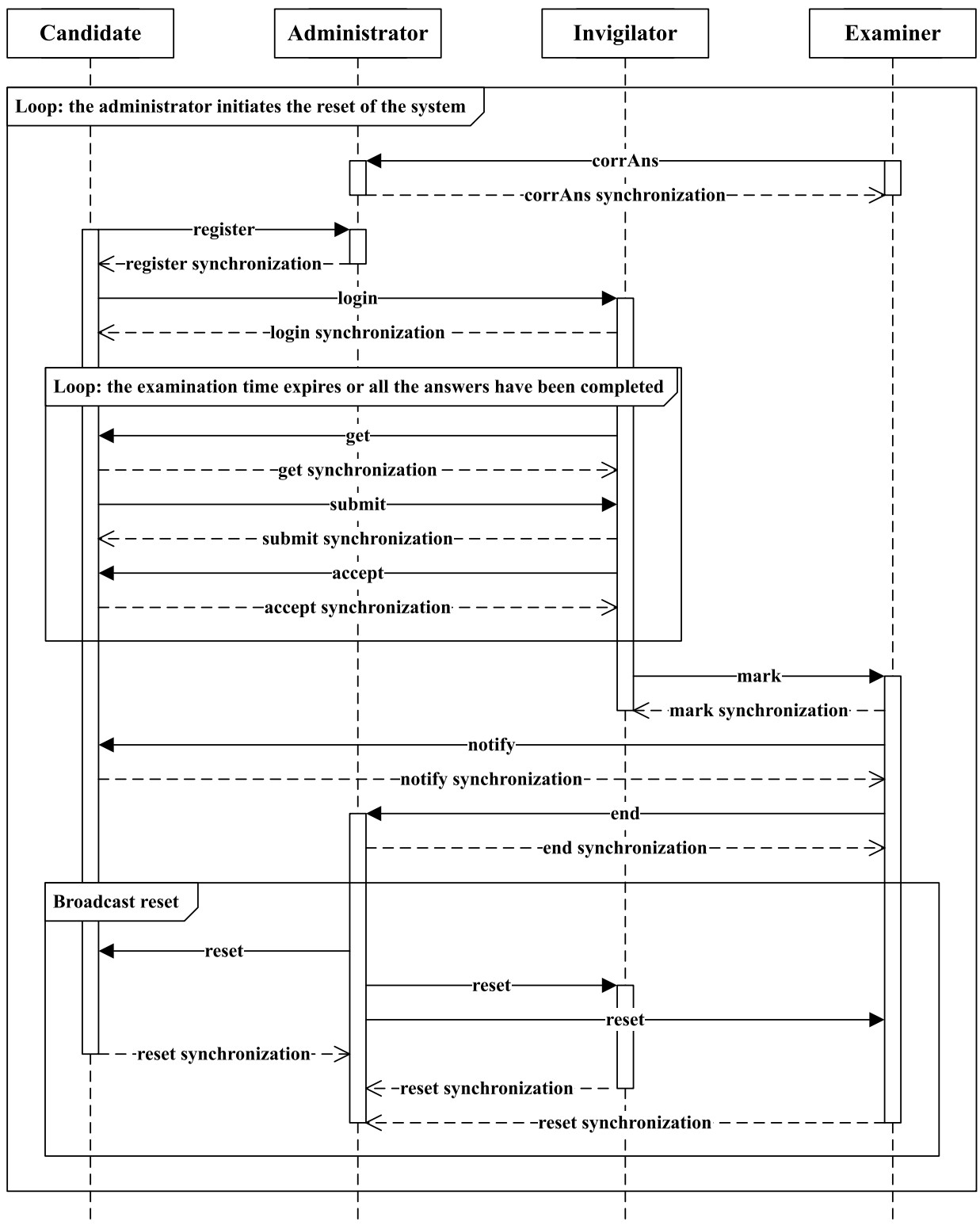

**Figure 1** Interactions among examination participants.

administrator sending a `reset` signal, which all other components receive and act upon, restoring the system to its initial state. There are two main loops in our sequence:

- *Candidate answering loop*: This loop involves the candidate answering questions one by one. Upon receiving each answer, the invigilator accepts the answer and moves on to the next question until the candidate completes all their answers or the candidate's examination time expires.
- *Examination conclusion loop*: This loop addresses the overall conclusion of the examination process. Once all candidates have been notified of their scores, the administrator initiates the reset of the system, ensuring that all processes are completed, and the system is prepared for the next examination session.

These loops help ensure that every response is systematically captured and evaluated, enabling the examination to conclude accurately and the system to transition smoothly to the next session.

### Model simplification and justification

The model simplification considered in this article includes multiple students, but only one examiner and one invigilator. This simplification is justified to some extent because it allows for a focused analysis of the interactions between different roles involved in electronic examinations while managing complexity.

*Focused analysis*. By reducing the number of examiners and invigilators to one, we can concentrate on the key interactions between candidates and the examination administration. This makes it easier to understand the overall system behaviors with a focus on candidate-centric interactions. We acknowledge that having multiple invigilators and examiners would be more realistic and provide a more generalizable model, but this may result in complex scheduling strategies, which is beyond the scope of this article's assumptions.

*Manageability and foundational understanding*. The simplified model is more manageable for several reasons. On the one hand, it reduces variables and interactions, making it easier to track and analyze processes. On the other hand, it clarifies core issues and interactions, resulting in a more straightforward and manageable system. This manageability provides a clear starting point for developing more complex models. This approach, while basic, helps identify fundamental issues and interactions crucial for further analysis, such as statistical model checking and temporal analysis.

## UPPAAL

UPPAAL, a real-time system model checker based on timed automata, developed collaboratively by Uppsala University and Aalborg University (*Behrmann, David & Larsen, 2004*; *David et al., 2015*; *Uppsala University & Aalborg University, 2023*), is widely used in domains such as network protocols (*Valero, Diaz & Cambronero, 2017*), transportation systems (*Basile et al., 2021*), and cyber-physical systems (*Hasrat et al., 2023*). UPPAAL comprises an editor, two simulators, and a verifier. The functions of the editor, simulator, and verifier are briefly introduced as follows.

### Editor

The editor is utilized for system modeling, involving the construction of a network of timed automata. A system model primarily includes declarations for global variables, templates (comprising local variable declarations), and system declarations. Global variables encompass synchronization variables, clock variables, and other relevant variables. Each template corresponds to a timed automaton and serves as a model specification, as detailed below. System declarations instantiate templates, leading to in the creation of a network of timed automata.

*Model specification*. The target system is represented as a network of timed automata using UPPAAL's graphical editor. Types of timed automata are established as process templates in UPPAAL, with each process template capable of process instantiation as a set of automata of the same type. A process template primarily consists of two elements: locations and edges (*Uppsala University & Aalborg University, 2023*).

*Location*. A location in UPPAAL represents a process state within the system. A process state represents a stay point in the execution of a process instance. Process states can have invariants, which are conditions that must be true for the system to stay at that state. There are four types of locations, which help define specific behaviors and timing constraints within the system, allowing for precise modeling and verification of real-time properties.

– *Regular location*: A state in the timed automaton where the system can stay and time can pass.

– *Initial location*: The starting state of a timed automaton.

– *Urgent location*: Time does not pass while the system is in this location.

– *Committed location*: Time also does not pass in this location; however, the next transition must be an outgoing edge from any committed location.

• *Edge*. An edge connects two locations (or itself), representing the transition between two states. It involves four types of expressions: select, guard, synchronization, and update.

– *Select*: The non-deterministic selection of a value within a range.

– *Guard:* A Boolean expression used to determine the enabling of a transition.

– *Synchronization*: Synchronization is achieved through channels, using two expressions, `c!` and `c?`, on the channel variable `c`. The `c!` expression sends a synchronization signal, while the `c?` expression receives it. Synchronization ensures that transitions in different automata occur simultaneously, allowing for coordinated behavior between concurrent processes.

– *Update*: Changing values of variables.

Variables in UPPAAL can be integers, Booleans, clocks, constants, arrays, structs, channels, or doubles. Integer and Boolean variables capture various aspects of the system, such as event counts or conditions, and can be read and updated during transitions. Clock variables manage time, increasing continuously and being reset as needed. They are used in invariants and guards to enforce timing constraints, ensuring transitions occur within specific time bounds.

Channel is a synchronization mechanism used in timed automata to facilitate communication between different automata or processes. Channels enable processes to synchronize their transitions by sending and receiving signals. Channels in UPPAAL are used as follows:

- `channel c`: This declares a specific channel named `c` used for communication.
- `c!`: This denotes a sending action on channel c. When a process executes `c!`, it sends a signal through channel c, indicating that it is ready to synchronize with any process that is set to receive on this channel.
- `c?`: This denotes a receiving action on channel c. When a process executes `c?`, it waits to receive a signal through channel `c` and will synchronize its transition with the process sending the signal.

There are two types of channels in UPPAAL:

- *Binary channels*: These channels enable one-to-one communication between two processes. One process will send a signal (`c!`) while the other process will receive it (`c?`), and both processes will synchronize their transitions simultaneously.
- *Broadcast channels*: These channels allow a signal to be sent to multiple receivers simultaneously. Any process set to receive the signal will synchronize with the sender.

Time management in UPPAAL is achieved with clocks, which define timing constraints and conditions for locations and transitions. In our UPPAAL model, we use two clocks, `t` and `pt`. The clock `t` manages the candidates' examination time, specifically during the `get`, `submit`, and `accept` loop, where the answering process ends when `t` exceeds a threshold value, `MaxT`. The clock `pt` manages the entire examination period, covering activities such as `corrAns`, `register`, `login`, `mark`, *etc*. This time management is implemented by associating clock conditions with channel synchronization, ensuring that synchronization occurs when the guard conditions involving the clocks are satisfied, and by assigning invariants to locations to ensure that the process can stay at a location only if the invariant conditions are met.

Information management in UPPAAL involves updating and synchronizing variables across different components of the system. Transitions can assign new values to variables, and channels can be used to synchronize information between concurrent processes. For example, during a transition, a process might synchronize with another process using a channel and update a variable to reflect the occurrence of an event. In our UPPAAL model, we use several queues to record variable information (such as candidate id, question id, or answer id) and synchronization information (such as operation type). This method aims to facilitate further verification processes.

### Simulator

The simulator serves the purpose of (1) validating and debugging the model *via* interactive and visual step-through behavior, and (2) demonstrating (counter) example/witness traces/paths. Two types of simulators are available: a symbolic simulator and a concrete

simulator. The symbolic simulator executes "symbolic" transitions, which correspond to many compressed concrete transitions. While this simulator is very effective, it can be confusing. In contrast, the concrete simulator uses concrete timings, making it easier to understand but less effective. Both simulators are used for trace displays: the symbolic simulator is used for symbolic queries, and the concrete simulator is used for SMC queries. These simulators offer a variety of operations, including *Reset*, *Next*, *Prev*, *Replay*, *Random*, *Shrink*, *Expand*, and *Speed Selection*. Within the graphical windows, state diagrams and message sequence charts are presented to enhance the understanding of simulations.

### Verifier

The verifier exhaustively explores the state space corresponding to a system model, checking whether the specified properties are satisfied. The properties are specified using a specific query language.

*Property specification.* In UPPAAL, the query language is a subset of timed computation tree logic (TCTL) language (*Alur, 1992*). The query language primarily encompasses five types of path formulae, *i.e.*, `E<>p`, `A<>p`, `E[]p`, `A[]p`, and `p->q`, where `p` and `q` are state formulae such as `i==4`. In UPPAAL, state formulae are logical expressions used to specify and verify properties of system states. They involve variables, clocks, and state conditions to check requirements such as safety, liveness, and reachability. For example, a state formula might ensure that a variable never exceeds a certain value or that a specific state is eventually reached.

`E<>p` signifies there exists a path with a state satisfying the predicate `p`. Note that `E` quantifies over paths, while `<>` quantifies over a state in a path. `A<>p` indicates that in every path, there exists a state that satisfies `p`. `E[]p` denotes that there exists a path where all states satisfy `p`. `A[]p` signifies that in all paths, all states satisfy `p`. Finally, `p->q` means that whenever `p` is satisfied, `q` is also satisfied in future.

Once users define property specifications and click the *Check* button, the verifier checks whether the system model satisfies each property.

## MODELING ELECTRONIC EXAMINATION USING UPPAAL

In this section, we provide a detailed introduction to the notation, model specification, and property specification related to electronic examinations.

### Notation

There are primarily three data structures: *Operation*, *Item*, and *TotalScore*. We present their definitions as follows. An operation is a quadruple (*id*, *op*, *q*, *a*) where:

1) *id* represents the identifier of a candidate, where $id \in \mathbb{Z}$.

2) *op* denotes an operation, where $op \in$ {*register*, *login*, *get*, *submit*, *accept*, *corrAns*, *mark*, *notify*, *end*, *reset*}.

3) *q* indicates a question.

4) *a* indicates an answer.

An operation captures the details of an action performed by a role, encompassing four types of information: candidate identifier, operation, question, and answer. Each candidate has a unique identifier. Eleven operations are considered to facilitate interactions among different roles. The *mark*, *end*, and *reset* indicate distinct phases in an electronic examination. For example, *mark* indicates that the examination has concluded, and it is time for scoring. *register* and *login* require only candidate identifier information. *corrAns* is used to set the correct answer for each question, necessitating information about the question. Similarly, *notify* requires candidate identifier information. Finally, *get*, *submit*, and *accept* are designated for handling actions such as receiving a question, submitting an answer, and accepting a submission. An example of a submission operation is provided in Example 3.1.

*Example 3.1.* A submission operation, denoted as $(cand_1, submit, q_2, a_2)$, signifies that the candidate $cand_1$ is submitting an answer $a_2$ in response to question $q_2$.

In the UPPAAL implementation, when certain information is unnecessary, we set the corresponding value to −1. An example of a correct answer setting operation is illustrated below.

*Example 3.2.* A correct answer setting operation is denoted as $(-1, corrAns, q_3, a_3)$. This operation indicates the correct answer for a question $q_3$ is set to $a_3$.

For each question, the candidate submits a corresponding answer, and the examiner marks it according to the correct answer. We define an *item* as a tuple that associates a question with the candidate's answer and the score marked by the examiner. An *item* is a triple $(q, a, s)$ where:

1) $q$ represents a question.

2) $a$ represents an answer.

3) $s$ denotes the score corresponding to the question and the answer.

An *item* exclusively records information about a single question. Consequently, we further define *MarkScore* to collect all the items and calculate the total score to notify a candidate. A *candidate* is a pair (*items*, *total*) where:

1) *items* indicates a set of items.

2) *total* denotes the total score with respect to the items.

After defining the above concepts and encoding them into declarations in UPPAAL, we can construct a series of models for an electronic examination system using these data structures.

## Model specification

In UPPAAL, the model specification is presented in the form of timed automata templates. In our model, there are four templates corresponding to the candidate, administrator, invigilator, and examiner, as follows.

### Candidate template

A candidate is an individual taking the examination as shown in Fig. 2. Initially, the candidate must register with the administrator using the `register[i]!` synchronization.

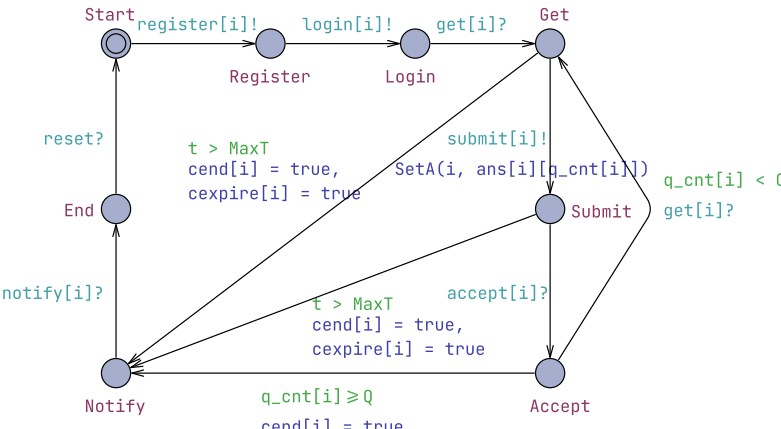

**Figure 2 Candidate template.**

Following a permission check, the candidate can log in to commence the examination. The current question number, denoted as `q_cnt`, is initialized to 0. The candidate engages in a loop where they receive a question, submit its answer, and confirm the acceptance of the answer.

The `get[i]?` synchronization is used to acquire questions sequentially from the invigilator. Upon obtaining a question, the question number is recorded in `cand`, a data structure implementing a set of *candidate*s. After receiving to the question, the candidate uploads the answer using the `submit[i]!` synchronization.

When there is a submission, the synchronization occurs using `Submit[i]!` and then the function `SetA(i, ans[i][q_cnt[i]])` is executed. Due to UPPAAL not directly supporting string types, integers are used to denote answers. The Candidate template is parameterized to directly set the candidates' answers.

```
//set a current answer of candidate i
SetA(ID i, ANS a){
        cand[i].item[q_cnt[i]].a = a;
}
```

The `SetA(ID i, ANS a)` function assigns the answer `a` (which is set when a `Candidate` template is instantiated) to the current question item of the candidate identified by `i`. Specifically, it updates the answer field `a` in the `item` structure for the candidate `i` at their current question count `q_cnt[i]`. This helps in managing and recording the candidates' answers during an electronic examination.

To ensure acknowledgment of the answer by the invigilator, the candidate uses an `accept[i]?` synchronization. When the examination time expires (`t > MaxT`) or the candidate has completed all their answers (*i.e.*, the number of answered questions equals or exceeds `Q`, where `Q` is the total number of questions), the candidate transitions to the `Notify` location to await their scores, and the `cend[i]` flag is set to true. Finally, upon receiving notification of their scores through the `notify[i]?` synchronization, the candidate reaches the `End` location. Subsequently, all states are reset using the `reset?`

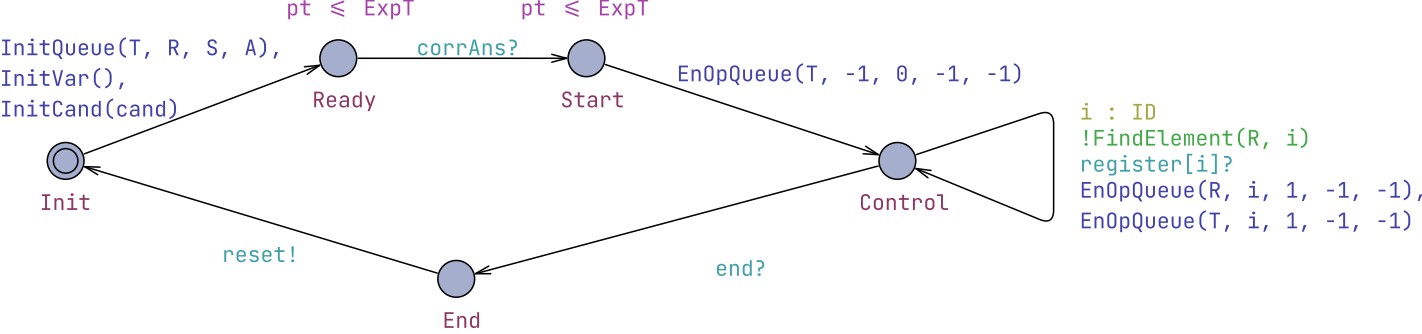

**Figure 3 Administrator template.**

synchronization. The examination process ends when all candidates have been notified of their scores, as explained in the "Preliminaries" section.

Upon reaching the End location, we use `reset?` to reinitialize a candidate. The `reset` is a broadcast channel. All candidates, the invigilator, and the examiner can only be reset after the `Administrator` sends a `reset!` signal. This ensures that the system is restored to its initial state.

### Administrator template

The administrator is responsible for information management, encompassing initialization, correct answer setting, and registration information maintenance, as shown in Fig. 3. Before any processes, the administrator automaton must initialize all necessary variables and reach the `Ready` location. Through the `corrAns` synchronization, correct answers to questions are set. Subsequently, the administrator automaton moves to the `Start` location, signifying the commencement of the examination. Concurrently, a start operation, encoded as (-1, 0, -1, -1), is added to the T queue. Note that the entire examination time is controlled by the state invariant (pt ≤ ExpT).

In our system, queues are used to manage various operations related to candidates during the electronic examination process. Four instances of `OpQueue` are declared to categorize different types of operations: `T` for total operations, `R` for register operations, `S` for submit operations, and `A` for accept operations. Each queue stores operations that are performed during the examination, such as registering candidates, submitting answers, and accepting results. These structures and instances collectively facilitate the organized and efficient processing of tasks within the system.

```
//T: Total, R: Register, S: Submit, A: Accept
OpQueue T, R, S, A;
```

The `OpQueue` structure represents these queues and includes fields for storing operations and managing the front and rear of the queue. The `OpQueue` structure manages a queue of `Operation` elements, using an array (`Operation data[MaxSize]`) to store the elements of the queue, with `front` and `rear` indices indicating the head and tail of the queue, respectively. The `OpQueue` structure is defined as follows:

```
//OpQueue Type
typedef struct{
        Operation data[MaxSize]; //store the elements of a queue
        int front; //queue head
        int rear; //queue tail
}OpQueue;
```

Each `OpQueue` can store multiple operations, where each operation is represented by the `Operation` structure. The fields in `Operation` include `id` for the identifier of a candidate, `op` for the type of operation (*e.g.*, `register`, `submit`), `q` for the question identifier, and `a` for the answer provided by the candidate. The operation type is encoded by integer values, such as −1 for none, 0 for `start`, 1 for `register`, 2 for `login`, and so on, up to 10 for `reset`. Note that the use of numeric values for operation types is necessary because UPPAAL does not support string types.

```
//Operation Type
typedef struct{
        NCID id;    //candidate id
        OpType op;//operation type: -1: none, 0: start, 1: register,
                                    2: login, 3: get, 4: submit,
                                    5: accept, 6: corrAns, 7: mark,
                                    8: notify, 9: end, 10: reset
        NCQS q;        //question id
        ANS a;         //answer id
}Operation;
```

We initialize and manage these queues using various functions. Their declarations are as follows and more information about implementations can be found in the GitHub site (*Zhou, 2024*). We briefly introduce their functionality here. The `InitOpQueue` function initializes a queue by setting all operation fields to default values and resetting the front and rear indices. The `EnOpQueue` function adds a new operation to the queue. The `FindElement` function determines whether an operation with candidate identifier `xi` belongs to the `Q` queue. The `FindOperation` function determines whether an operation belongs to the `Q` queue based on all operation fields.

```
//Initialize an OpQueue
void InitOpQueue(OpQueue& q)

//Add an element to an OpQueue
bool EnOpQueue(OpQueue& q, int xi, int xop, NCQS xq, int xa)

//Find an element in an OpQueue according to identifier
bool FindElement(OpQueue q, int xi)

//Find an element in an OpQueue according to all operation fields
bool FindOperation(OpQueue q, int xi, int xop, int xq, int xa)
```

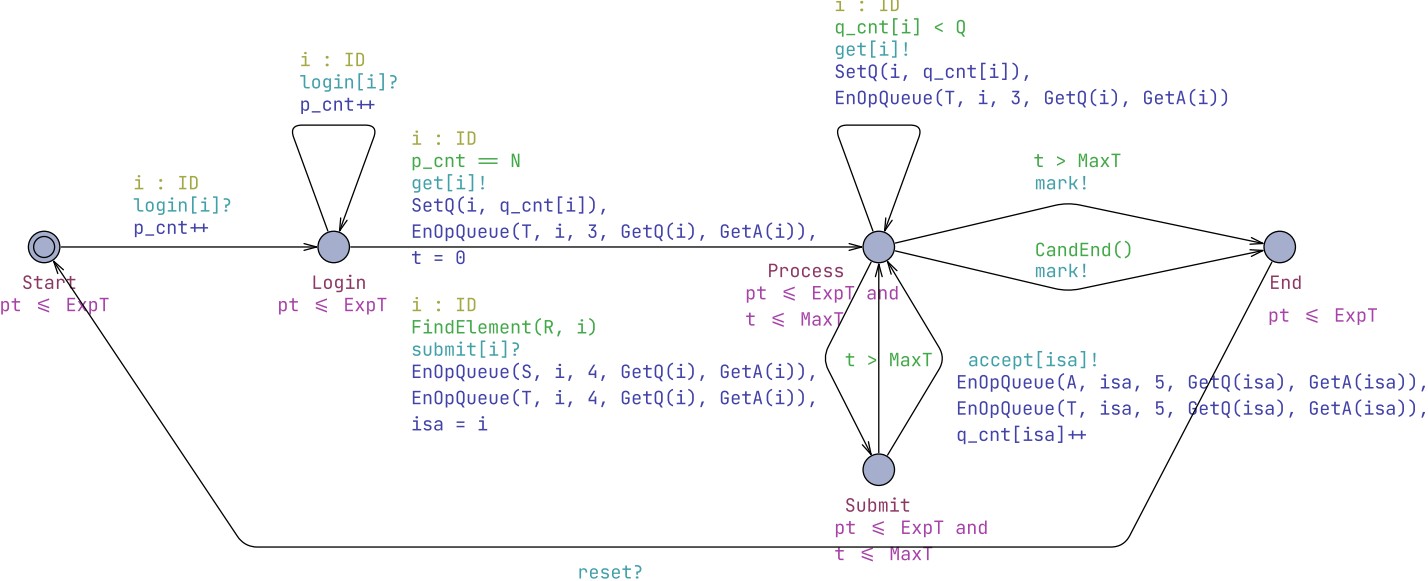

**Figure 4 Invigilator template.**                               

When a candidate requests registration in the examination system, the administrator checks whether the candidate has already registered. If not, the candidate's registration information is stored in the R queue. Additionally, a `register` operation, encoded as `(i, 1, -1, -1)`, is enqueued to the T queue. The `(i, 1, -1, -1)` tuple encodes that candidate i has been registered, with 1 indicating the `regist` operation. The question and answer fields are set to -1, denoting "none", as this information is not required for the registration process. Finally, when the examiner confirms that all automata have reached an End location through the `end` synchronization, the administrator issues an instruction to reset all automata.

### Invigilator template

An invigilator oversees the examination proceedings, taking on the responsibility of checking candidates' logins, dispatching questions to candidates, receiving and confirming the submission of answers, as shown in Fig. 4.

Initially, as candidates log in, the system transitions to the Login location. Once all candidates are prepared for the examination (*i.e.*, the number of logged-in candidates equals N), the invigilator synchronizes with candidates using `get[i]!` and `SetQ` to transmit and update the questions. Simultaneously, a `get` operation, represented as `(i, 3, GetQ(i), GetA(i))`, is placed into the T queue.

Upon a candidate submitting an answer, the invigilator checks their registration. If confirmed, the `submit[i]?` synchronization takes place, and a `submit` operation, encoded as `(i, 4, GetQ(i), GetA(i))`, is queued. The local variable `isa` is used to store the current candidate id to avoid any potential confusion. Subsequently, the automaton reaches the Submit location. Through the `accept[isa]!` synchronization, the invigilator

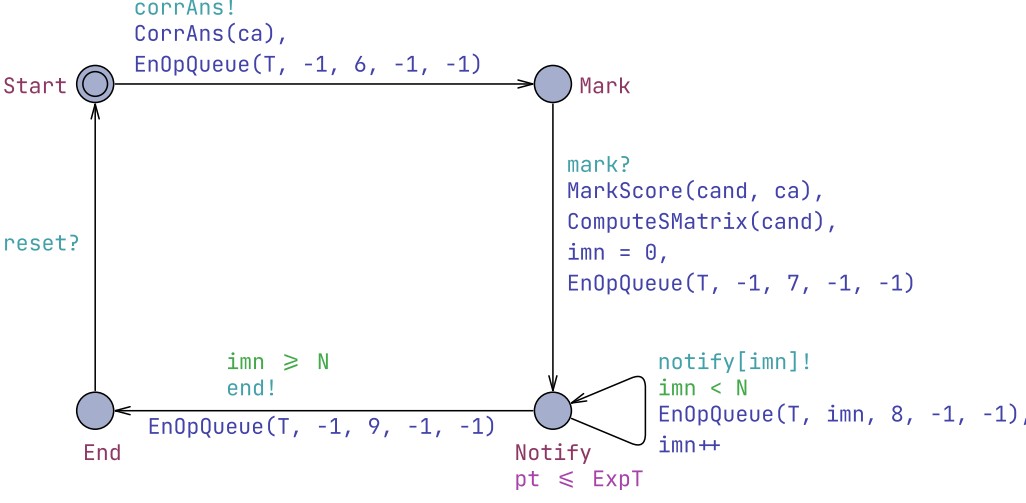

**Figure 5** Examiner template.

notifies the candidate that their answer to the question has been received. Additionally, the `accept` operation is added to the `T` queue, and the question counter is updated.

Finally, if all candidates have responded to all questions, the invigilator automaton advances to the `End` location, notifies the examiner that the marking process can commence, and sets the `iend` flag to true. Note that if there is a timeout (`t > MaxT`), regardless of whether it occurs at the `Submit` or `Process` location, the system will transition to the `End` location immediately.

### Examiner template

An examiner is responsible for setting the correct answers to questions, evaluating candidates' responses, and communicating their scores to the candidates, as shown in Fig. 5. Initially, the examiner employs the `corrAns` synchronization to provide the correct answers before the examination, and a `corrAns` operation, represented as (−1, 6, −1, −1), is inserted into the `T` queue. In the meantime, the `CorrAns` function is executed, which sets the correct answers for the questions. The `ca[Q]` is an array of type `CAS`, where `CAS` is of type `int`. This means that each question has a correct answer. Due to the array's characteristics, the identifiers of the `Q` questions range from 0 to `Q−1`. For simplicity, we assume that each correct answer is set to the question's identifier, as our purpose is to analyze the model rather than use real information. However, the correct answer array can be set to any integer values.

```
//correct answers type
typedef int CAS;

//the question with id belongs to [0, Q-1] having a correct answer
CAS ca[Q];

//Set correct answers ca[Q] to questions
void CorrAns(CAS& ca[Q]){
```

```
        int i;
        for(i = 0; i < Q; i++)
            ca[i] = i;
}
```

Subsequently, upon receiving the synchronization signal `mark?`, the examiner assesses each candidate. A `mark` operation, encoded as (−1, 7, −1, −1), is then added to the T queue. The marking process is achieved by executing the `MarkScore` function. The `MarkScore` function is used to evaluate and assign scores to each candidate based on their answers. The function iterates through all candidates and their respective answers, comparing each candidate's answer with the correct answer stored in the `ca` array. If a candidate's answer matches the correct answer, the score (`cand[i].item[j].s`, where `i` denotes the candidate identifier and `j` denotes the question identifier) for that question is set to 1; otherwise, it is set to 0. The function also calculates the total score (`cand[i].total`) for each candidate by summing the scores of all their answers.

```
//mark socres and the total score for every candidate
void MarkScore(CandidateType& cand[N], CAS& ca[Q]){
        int i, j, k;
        for(i = 0; i < N; i++)
        {
            cand[i].total = 0;
            for(j = 0; j < Q; j++)
            {
                if(cand[i].item[j].a == ca[j])
                    cand[i].item[j].s = 1;
                else
                    cand[i].item[j].s = 0;
                cand[i].total = cand[i].total + cand[i].item[j].s;
            }
        }
}
```

Note that the `cand` variable represents an array of type `CandidateType`, corresponding to *candidate* introduced in "Preliminaries". The `CandidateType` type comprises two elements: `item[Q]` and `total`. The `item` array is of type `ItemType`, which includes the question (q), the answer (a), and the score (s). The `total` element represents the sum of scores for all questions answered by the candidate.

```
//Candidate Type
typedef struct{
        ItemType item[Q]; //items
        int total; //total score: sum(s[Q])
}CandidateType;
```

```
CandidateType cand[N];
//Item Type
typedef struct{
        NCQS q; //question
        ANS a; //answer
        MS s; //score
}ItemType;
```

The `ComputeSMatrix` function is executed simultaneously with the `mark` synchronization. This function computes the similarity of answers between candidates for subsequent cheater detection. We consider a simplified version of cheater detection by referring to the approach described in *Kassem, Falcone & Lafourcade (2017)*. In our article, if two candidates have a high number of the same answers, it may indicate a cheating event. The `ComputeSMatrix` function computes the similarity matrix for a set of candidates. It compares the answers of each pair of candidates for all questions. If two candidates have the same answer for a question, the similarity score for that pair is incremented. This results in a matrix (`sm`) where each element `sm[i][j]` represents the number of questions for which candidates `i` and `j` provided the same answer.

```
//compute Similarity Matrix
void ComputeSMatrix(CandidateType& cand[N]){
        int i, j, k;
        for(i = 0; i < N; i++)
            for(j = i + 1; j < N; j++)
                for(k = 0; k < Q; k++)
                    if(cand[i].item[k].a == cand[j].item[k].a)
                        sm[i][j]++;
}
```

Following the evaluation, the examiner uses the `notify[imn]!` synchronization to inform candidates of their scores. Finally, all candidates are notified of their scores (*i.e.*, `imn` ≥ N).

### System declaration

Based on the four UPPAAL templates mentioned above, a comprehensive system can be generated by interconnecting these timed automata into a network. The system declaration is presented as follows. The following code snippet initializes the main components of the electronic examination system. `C0` and `C1` represent two candidates, initialized with the identifiers `0` and `1`, respectively. Assume there are three questions, with the answers for `C0` and `C1` set to {0, 1, 2} and {2, 1, 0}, respectively. `AD` is the administrator, `I` is the invigilator, and `E` is the examiner. The system statement combines these components into a single system, enabling interaction between candidates, the administrator, the invigilator, and the examiner during the electronic examination process. Here, we have instantiated two candidates and additional candidates can be added in a similar manner.

```
//Template instantiations
const array answer = {{0, 1, 2}, //candidates' answers
                      {2, 1, 0}};
C0 = Candidate(0, answer);
C1 = Candidate(1, answer);
AD = Administrator();
I = Invigilator();
E = Examiner();
//System Declaration
system C0, C1, AD, I, E;
```

## Property specifications

A trustworthy electronic examination system must adhere to a set of properties to ensure reliability. These properties can be articulated through formal specifications. According to the literature (*Kassem, Falcone & Lafourcade, 2017*), we consider the following twelve properties, which are encoded using a simplified version of TCTL, serving as property specifications in UPPAAL.

*(1) No deadlock*

In the electronic examination model, the absence of deadlocks is crucial to prevent any "never-ending" scenarios. Deadlocks, where processes wait indefinitely for each other, disrupt the system's flow. By implementing effective process synchronization and careful system design, we ensure a smooth examination experience, free from any prolonged or unresolved situations. Describing the absence of deadlocks in UPPAAL as a query is straightforward:

```
A[] not deadlock
```

*(2) Candidate registration*

The candidate registration property requires that a candidate can submit an answer only if they have registered. This assertion is verified using two queues, namely `R` and `S`, dedicated to storing `register` and `submit` operations, respectively. The query is formulated as follows, where `forall(i:ID)` denotes every candidate, and `FindElement (Q, i)` is a function that determines whether an operation with candidate identifier `i` belongs to the `Q` queue. The candidate registration property emphasizes that submission is contingent upon prior registration, and this condition is rigorously checked through the `R` and `S` queues.

```
A[] forall (i:ID) not (not FindElement(R,i) and
                    FindElement(S,i))
```

The query asserts that for every state within each path of the state space, there is no instance where an operation exists in the `S` queue but is absent in the `R` queue. In simpler terms, it implies that no candidate can submit without first registering.

### (3) Candidate eligibility

The candidate eligibility property signifies that a candidate's answer can be accepted only if they have registered. To formulate this query, we use two queues `R` and `A`, specifically for the `register` and `accept` operations. The query is expressed as follows.

```
A[] forall (i:ID) not (not FindElement(R,i) and
                       FindElement(A,i))
```

In this query, for every state within each path of the state space, there is no instance where an operation exists in the `A` queue but is absent in the `R` queue. In other words, it asserts that if a candidate does not register, their answer cannot be accepted.

### (4) Answer authentication

The answer authentication property requires that a candidate's answer can be accepted only if they have submitted the answer. To articulate this query, we use two queues for `submit` and `accept` operations, denoted as `S` and `A` respectively. The query is formulated as follows.

```
A[] forall (i:ID) not (not FindElement(S,i) and
                       FindElement(A,i))
```

This query asserts that, for every state within each path of the state space, there is no instance where an operation exists in the `A` queue but is absent in the `S` queue. In essence, it emphasizes that if a candidate does not submit an answer, that answer cannot be accepted.

### (5) Answer singularity

The answer singularity property signifies that, for each candidate, only a singular response can be deemed acceptable per question. The function `OneAnswerEachQuestion` is designed to verify the presence of operations within queue A that share identical questions.

```
A[] forall(i:ID) OneAnswerEachQuestion(A, i)
```

The query checks each state within every path of the state space, ensuring that the `OneAnswerEachQuestion` function consistently yields true. This indicates that, for each question, only a single response is admissible from a given candidate. The `OneAnswerEachQuestion` function is implemented as follows.

```
//check whether only one answer for every question xq from a candidate xi is
accepted in an OpQueue
bool OneAnswerEachQuestion(OpQueue q, int xi)
{
        int xq;
        int i, j;
        for(xq = 1; xq < Q + 1; xq++)
              for(i = q.front; i < q.rear; i++)
                    for(j = i + 1; j < q.rear; j++)
```

```
                        if(q.data[i].id == xi && q.data[j].id == xi && q.
data[i].q == xq  &&  q.data[j].q == xq)
                                return false;
        return true;
}
```

The `OneAnswerEachQuestion` function checks whether each candidate has provided only one answer per question in an `OpQueue`. It iterates through the queue, and for each question, it verifies that no candidate has submitted more than one answer. If a candidate is found to have multiple answers for the same question, the function returns false; otherwise, it returns true.

### (6) Acceptance assurance

The acceptance assurance property specifies the requirement that an answer submitted by a candidate should be accepted if the candidate's examination time has not expired. In this context, emphasis is placed on the initial submission, implying that the first `submit` operation related to a question from a candidate is succeeded by an accept operation. The following `FirstSubmitFollowAccept` function describes this query.

```
A[] forall(i:ID) FirstSubmitFollowAccept(T, i)
```

In this query, for each state in every path of the state space, the `FirstSubmitFollowAccept` function consistently yields true. This implies that, after the first submission of an answer from a candidate, the invigilator accepts the answer. The `FirstSubmitFollowAccept` function is implemented as follows.

```
//check whether the first occurrence of submit is followed by accept from a
candidate xi is accepted in an OpQueue if the candidate's examination time
has not expired
bool FirstSubmitFollowAccept(OpQueue q, int xi)
{
        int i;
        OpQueue tmp; //tmp records the first "submit" of a candidate and a
question
        InitOpQueue(tmp);
        for(i = q.front; i < q.rear + 1; i++)
  {
            if(q.data[i].id == xi && q.data[i].op == 4 &&
!FindOperation(tmp, xi, 4, q.data[i].q, -1) && i < q.rear && cexpire[xi] !=
true) //"!FindOperation" checks whether it is the first "submit" of
candidate xi
              {
                  if((q.data[i+1].id != xi || q.data[i+1].op != 5 || q.
data[i+1].q != q.data[i].q) && (q.data[i+1].id != -1 && q.data[i+1].op !=
```

```
            -1 && q.data[i+1].q != -1)) //except the sequence "{4,2,2,3},{-1,-1,-1,
      -1}", where {-1,-1,-1,-1} is a default operation
                                              return false;
                      EnOpQueue(tmp, xi, 4, q.data[i].q, -1);
                  }
              }
          return true;
      }
```

The `FirstSubmitFollowAccept` function checks whether the first occurrence of a
`submit` operation by a candidate is immediately followed by an `accept` operation for the
same question in an operation queue (`OpQueue`). It initializes a temporary queue to keep
track of the first `submit` operations for each question by the candidate. As it iterates
through the main queue, it verifies that each `submit` operation is followed by a
corresponding `accept` operation. Note that the `cexpire[xi] != true` expression is used
to exclude the scenario where the candidate's examination time has expired, as this
property does not apply in such cases. If any `submit` operation is not followed by an
`accept` operation, the function returns false. If all `submit` operations are followed by
`accept` operations, the function returns true.

### (7) Question ordering

The question ordering property emphasizes that a candidate can proceed to the next
question only after the answer to the current question is accepted. The `GetAcceptGet`
function formulates this property by specifying that a `get` operation for question `i` is
succeeded by an `accept` operation for the answer to question `i`. Furthermore, the `accept`
operation is succeeded by a `get` operation for question `i+1`. It is important to note that the
handling of the last question involves a special consideration.

`A[] forall(i:ID) GetAcceptGet(T, i)`

The aforementioned query signifies that, for each state in every path of the state space,
the sequence of operations, namely `get(i)-accept(i)-get(i+1)`, remains unbroken.
Consequently, this ensures that a candidate can systematically answer questions one after
another. The `GetAcceptGet` function is implemented as follows.

```
//check whether get(xi, q) is followed by accept(xi, q) and accept(xi, q) is
followed by get(xi, q + 1) in an OpQueue
bool GetAcceptGet(OpQueue q, int xi)
{
        int i, j, k;
        bool flag1 = 0;
        bool flag2 = 0;
        for(i = q.front; i < q.rear - 1; i++)
        {
                if(q.data[i].id == xi && q.data[i].op == 3)
```

```
                     {
                         flag1 = 0; //ensure that it is the first accept(xi, q) after
get(xi, q).
                         for(j = i + 1; j < q.rear; j++)
                         {
                             if(q.data[j].id == xi && q.data[j].op == 5 && flag1 ==
0)
                             {
                                 if(q.data[j].q != q.data[i].q)
                                     return false;
                                 flag1 = 1; //find the first accept(xi, q) after get
(xi, q), set flag1 to 1.
                                 flag2 = 0; //ensure that it is the first get(xi, q+1)
after accept(xi, q).
                                 for(k = j + 1; k < q.rear + 1; k++)
                                 {
                                     if(q.data[k].id == xi && q.data[k].op == 3 &&
flag2 == 0)
                                     {
                                         if(q.data[k].q != q.data[i].q + 1)
                                             return false;
                                         flag2 = 1; //find the first get(xi, q+1) after
accept(xi, q), set flag2 to 1. For the OpQueue {5, 2, 1, 0}, ... , {3, 2, 2,
0}, ... , {3, 2, 3, 0}, the {3, 2, 3, 0} should not be considered.
                                     }
                                 }
                             }
                         }
                     }
                 }
             }
         }
     return true;
}
```

The `GetAcceptGet` function checks the sequence of operations for a candidate in an operation queue (`OpQueue`). Specifically, it verifies whether a `get` operation (`get(xi, q)`) is followed by an `accept` operation (`accept(xi, q)`) and then by another `get` operation (`get(xi, q + 1)`) for the same candidate. The function iterates through the queue, setting flags to ensure that it correctly identifies and validates these sequences. If all sequences are correctly followed, the function returns true; otherwise, it returns false.

*(8) Exam availability*

The exam availability property requires that the acceptance of an answer from a candidate is permissible only during the examination period. This implies the presence of

an `accept` operation between the `start` operation and the end operation within the T queue. This property is specified using the `StartAcceptEnd` function in the following query.

```
A[] StartAcceptEnd(T)
```

The query checks whether the `StartAcceptEnd` function is satisfied for every state in every path of the state space. Given the singular occurrence of both the `Start` and `End` operations, we abstractly take the sequence spanning from `Start` to `End` as the examination period. The `StartAcceptEnd` function is implemented as follows.

```
//check whether every accept is preceded by start and followed by end in an
OpQueue
bool StartAcceptEnd(OpQueue q)
{
        int i, j;
        for(i = q.front; i < q.rear; i++)
        {
            if(q.data[i].op == 5)
                for(j = i + 1; j < q.rear + 1; j++)
                    if(q.data[i].op == 0)
                        return false;
            if(q.data[i].op == 9)
                for(j = i + 1; j < q.rear + 1; j++)
                    if(q.data[j].op == 5)
                        return false;
        }
        return true;
}
```

The `StartAcceptEnd` function checks whether every `accept` operation (op == 5) in an `OpQueue` is preceded by a `start` operation (op == 0) and followed by an `end` operation (op == 9). It iterates through the queue to ensure that each `accept` operation has a preceding `start` operation and no subsequent `accept` operation after an `end` operation. If these conditions are met, the function returns true; otherwise, it returns false.

*(9) Answer-score integrity*

The answer-score integrity property ensures that the correct answer cannot be modified after the examination starts. This property is captured as the following `NoStartCorrAns` function in the query.

```
A[] NoStartCorrAns(T)
```

This query guarantees that no `corrAns` operation is succeeded by `start` in the T queue, signifying that the correct answers can only be set before the commencement of an examination. The `NoStartCorrAns` function is implemented as follows.

```
//check whether no corrAns is followed by start in an OpQueue
bool NoStartCorrAns(OpQueue q)
{
        int i, j;
        for(i = q.front; i < q.rear; i++)
              if(q.data[i].op == 0)
                    for(j = i + 1; j < q.rear + 1; j++)
                          if(q.data[j].op == 6)
                                return false;
        return true;
}
```

The `NoStartCorrAns` function checks whether there are no `start` operations (`op ==` 0) that follow a `corrAns` operation (`op == 6`) in an `OpQueue`. It iterates through the queue and ensures that if a `start` operation is encountered, no subsequent `corrAns` operations are found. If any `corrAns` operation is found after a `start` operation, the function returns false; otherwise, it returns true.

### (10) Cheater detection

During an examination process, cheating may take place, *e.g.*, one candidate copies the answers of the other candidate. In this article, we only consider the basic form of cheating, namely copying. The `NoDistanceExceed` function is employed to verify this property.

```
A[] NoDistanceExceed(sm)
```

To assess the answer similarity between two candidates, a matrix, denoted as `sm[i][j]`, is computed to measure the degree of similarity between candidate `i` and candidate `j`. If `sm[i][j] > D` where D is a constant representing the tolerance for duplication, the `NoDistanceExceed` function returns false, indicating a potential cheating event. Here, the `sm[i][j]` corresponds to the count of same answers between candidate `i` and candidate `j`. The `NoDistanceExceed` function is implemented as follows.

```
//check whether there exists a distance between two candidates is greater
than D
bool NoDistanceExceed(int sm[N][N])
{
        int i, j;
        for(i = 0; i < N; i++)
              for(j = 0; j < N; j++)
                    if(sm[i][j] > D)
                          return false;
        return true;
}
```

Here, we consider a simplified cheater detection method based on the approach described in *Kassem, Falcone & Lafourcade (2017)*. This method detects potential cheating

by evaluating the similarity in answers between two candidates. We parameterized the `Candidate` template to allow setting the candidates' answers. For instance, if there are three questions and the answers for candidates `C0` and `C1` are {0, 1, 2} and {0, 1, 2} respectively, their answer similarity is 3, as computed by the `ComputeSMatrix` function. If the minimum allowed distance between candidates' answers is 2, then the cheater detection property is not satisfied. Conversely, if the answers for candidates `C0` and `C1` are {0, 1, 2} and {2, 1, 0}, their answer similarity is 1, and the cheater detection property is satisfied (this case is shown in our verification results).

*(11) Marking correctness*.

The marking correctness property asserts that once marking has occurred, the correct answers cannot be modified. This property is verified by the following query, using a function named `NoCorrAnsMark`.

```
A[] NoCorrAnsMark(T)
```

This query ensures that no `corrAns` operation is succeeded by the `mark` operation in the `T` queue, preventing any modification of correct answers during the marking process. The `NoCorrAnsMark` function is implemented as follows.

```
//check whether no corrAns is followed by mark in an OpQueue
bool NoCorrAnsMark(OpQueue q)
{
        int i, j;
        for(i = q.front; i < q.rear; i++)
            if(q.data[i].op == 7)
                  for(j = i + 1; j < q.rear + 1; j++)
                        if(q.data[j].op == 6)
                              return false;
        return true;
}
```

The `NoCorrAnsMark` function checks whether no `corrAns` operation (op == 6) is followed by a `mark` operation (op == 7) in an `OpQueue`. It iterates through the queue to ensure that if a `mark` operation is found, there is no subsequent `corrAns` operation. If this condition is met for all `mark` operations, the function returns true; otherwise, it returns false.

*(12) Mark integrity*.

The mark integrity property ensures that each candidate receives notification after marking, and all answers from candidates are duly marked. The verification of this property is performed through the `MarkIntegrity` function as follows.

```
A[] MarkIntegrity(T)
```

The `MarkIntegrity` function consists of two subfunction, namely `NoNotifyMark` and `AllMarked`. `NoNotifyMark` assesses whether there is no `notify` operation preceding a

mark operation. Simultaneously, `AllMarked` determines whether all the answers have been appropriately marked. Through these checks, the integrity of the marking process is assured. The `MarkIntegrity` function is implemented as follows.

```
//check the integrity of the marking process
bool MarkIntegrity(OpQueue q)
{
        if(FindOperation(q, 9, -1, -1, -1))
              if(NoNotifyMark(q) && AllMarked())
                      return true;
              else
                      return false;
        else
              return true;
}
```

The `MarkIntegrity` function checks the integrity of the marking process in an `OpQueue`. It first checks if an `end` operation (op == 9) is present in the queue using the `FindOperation` function. If an end operation is found, it then verifies that no `notify` operation is followed by an unmarked answer (using `NoNotifyMark(q)`) and that all answers have been marked (using `AllMarked()`). If both conditions are met, the function returns true; otherwise, it returns false. If no `end` operation is found, the function returns true. The `NoNotifyMark` and `AllMarked` functions are implemented as follows.

```
//check whether no mark is followed by notify in an OpQueue
bool NoNotifyMark(OpQueue q)
{
        int i, j, k;
        int c;
        for(k = 0; k < N; k++)
            for(i = q.front; i < q.rear; i++)
                 if(q.data[i].op == 8 && q.data[i].id == k)
                 {
                     c = q.data[i].id;
                    for(j = i + 1; j < q.rear + 1; j++)
                         if(q.data[i].op == 7 && q.data[i].id == c)
                             return false;
                 }
        return true;
}
```

The `NoNotifyMark` function checks whether a `notify` operation (op == 8) for any candidate is not followed by a `mark` operation (op == 7) for the same candidate in an `OpQueue`. It iterates through the queue to ensure that after a `notify` operation is found for

a candidate, no subsequent `mark` operation exists for that candidate. If this condition is met for all `notify` operations, the function returns true; otherwise, it returns false.

```
//check whether all the answers are marked (i.e., score is not -1) for every
candidate
bool AllMarked()
{
        int i, j;
        for(i = 0; i < N; i++)
            for(j = 0; j < Q; j++)
                if(cand[i].item[j].s == -1)
                    return false;
        return true;
}
```

The `AllMarked` function checks if all answers for every candidate have been marked. It iterates through all candidates and their respective answers, returning false if any score is −1 (indicating the answer has not been marked). If all scores are valid (not −1), the function returns true.

## VALIDATION AND VERIFICATION IN UPPAAL

In our study, we use UPPAAL to rigorously verify the reliability of our electronic examination model. This includes evaluation metrics, verification results, and security aspects. Specially, the validation and verification process involved simulating a scenario and verifying key properties to ensure the model meets the required standards for an electronic examination system.

### Evaluation metrics

To evaluate the robustness and effectiveness of our electronic examination model, we used two evaluation metrics, focusing on functionality. These include:

- **Model consistency**: Ensuring that the model consistently adheres to a general examination process and validating that it aligns with common-sense understanding through simulations and manual checks.
- **Property verification**: Verifying key properties such as candidate registration, answer authentication, and exam availability, which are crucial for maintaining the reliability and integrity of electronic examinations.

While these metrics provide a comprehensive understanding of the model's reliability, there are certain limitations. The model is based on theoretical scenarios and simulations, which may not cover all real-world complexities. Performance under extreme conditions or with many candidates is yet to be fully assessed. These limitations highlight areas for future improvement and suggest further validation under diverse real-world conditions.

**Table 1 Parameter settings.**

| Parameter | Value | Meaning |
|-----------|-------|---------|
| N | 2 | The number of total candidates |
| Q | 3 | The number of total questions |
| M | 2 | The number of total scores |
| MaxSize | 50 | The max size of a queue |
| MaxT | 1,000 | The candidates' examination time |
| ExpT | 10,000 | The expiration period of the entire examination process (including setting correct answers, registration, and marking, *etc.*) |
| D | 2 | The minimum permissible distance between candidates' answers |
| U | 10 | The upper limit value of answers |

## Validation and verification results

To ensure the reliability of our electronic examination model, we conducted a verification process on the specified property specifications using UPPAAL. The verification experiment is performed on a computer equipped with an Intel(R) Core(TM) i7-1360P, 2,200 Mhz, running Java 17.0.7, and using UPPAAL version 5.0.0.

In our experiment, we focused on a scenario involving two candidates. It's noteworthy that more complex setups with additional candidates share similarities with this specific instance. The parameter configurations, detailed in Table 1, encompass a total of eight parameters. N is set to 2, indicating two candidates. Q is set to 3, denoting three questions. M is set to 2, signifying two scores. MaxSize is set to 50, indicating the maximum size of a queue is 50. MaxT is set to 1,000, denoting the maximum time (exam time) as 1,000. ExpT is set to 10,000, indicating the expiration period of the entire examination process (including setting correct answers, registration, and marking, *etc.*). Finally, the minimum permissible distance between candidates' answers is set to 2, denoted as D, and the upper limit value of answers is set to 10, denoted as U.

As shown in Fig. 6, we simulate the electronic examination in this scenario using UPPAAL and validate the model consistency of the examination process with colleagues. Based on their feedback, our model conforms to common-sense understanding of general electronic examination process.

Table 2 and Fig. 7 present the verification results of our electronic examination model using UPPAAL. Each property listed in the table corresponds to a specific aspect of the system's functionality that was verified. The results include the verification time, kernel time, total time, resident memory, and virtual memory peak for each property. The "Satisfied" result indicates that the property was successfully verified, meeting the specified criteria.

The properties verified include ensuring that the system does not reach a state where no further progress is possible (No deadlock); verifying that a candidate can only submit an answer if they have registered (Candidate registration); ensuring that only registered candidates can participate in the examination (Candidate eligibility); confirming that a candidate's answer is accepted only if it has been submitted (Answer authentication);

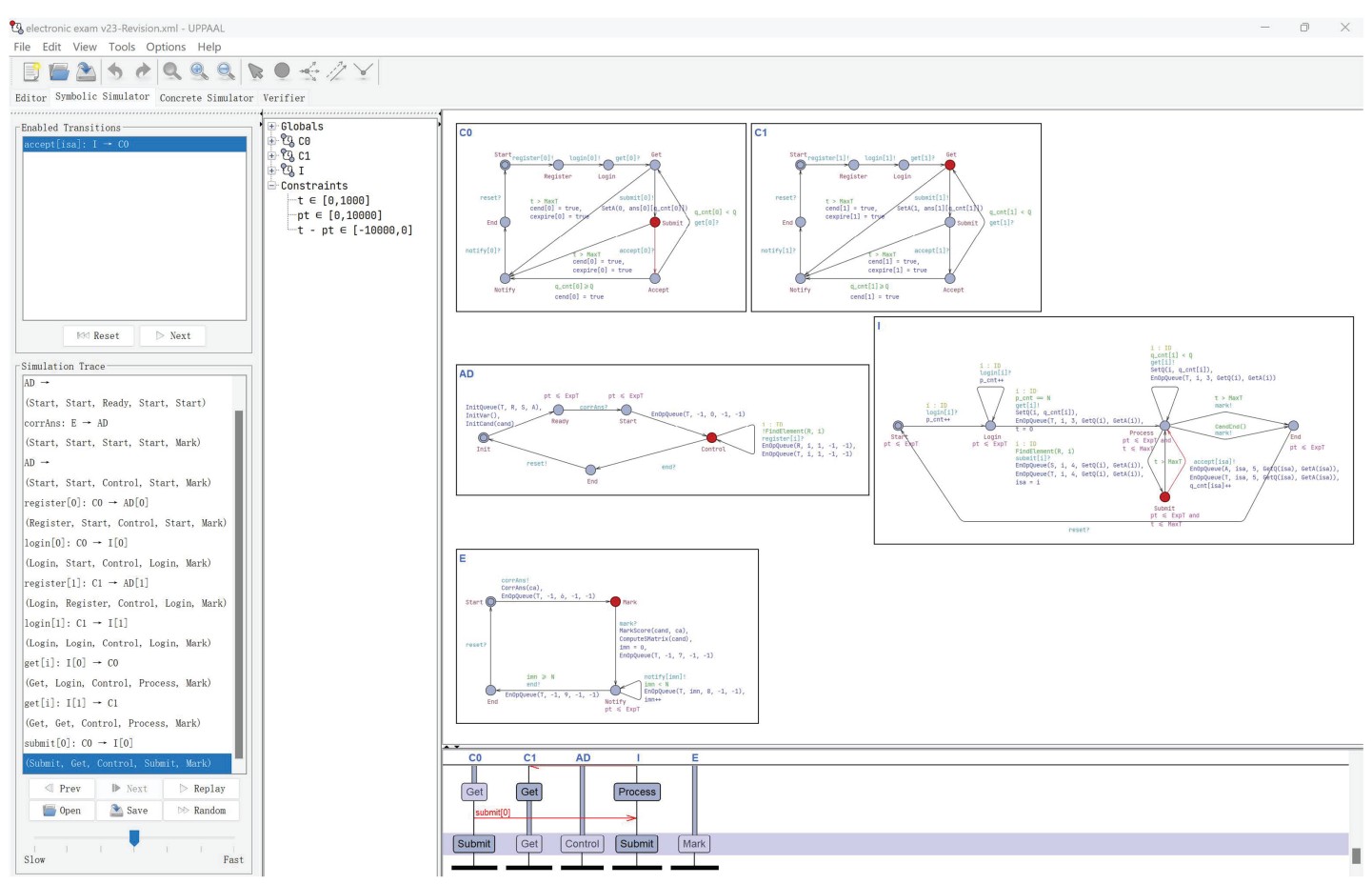

**Figure 6 Simulation and validation.**

**Table 2 Verification results.**

| No. | Property | Verification time/Kernel time/Total time (s) | Resident memory/Virtual memory peak (KB) | Result |
|---|---|---|---|---|
| (1) | No deadlock | 0.265 s/0 s/0.518 s | 39,760 KB/105,400 KB | Satisfied |
| (2) | Candidate registration | 0.032 s/0 s/0.398 s | 40,248 KB/106,356 KB | Satisfied |
| (3) | Candidate eligibility | 0.03l s/0 s/0.435 s | 39,972 KB/105,948 KB | Satisfied |
| (4) | Answer authentication | 0.047 s/0 s/0.43 s | 39,980 KB/105,964 KB | Satisfied |
| (5) | Answer singularity | 0.03l s/0.015 s/0.536 s | 39,892 KB/105,792 KB | Satisfied |
| (6) | Acceptance assurance | 0.344 s/0 s/0.892 s | 39,836 KB/105,708 KB | Satisfied |
| (7) | Questions ordering | 0.218 s/0 s/0.613 s | 39,868 KB/105,744 KB | Satisfied |
| (8) | Exam availability | 0.047 s/0 s/0.496 s | 39,896 KB/106,056 KB | Satisfied |
| (9) | Answer-score integrity | 0.094 s/0 s/0.407 s | 39,912 KB/106,076 KB | Satisfied |
| (10) | Cheater detection | 0.188 s/0 s/0.347 s | 39,924 KB/105,908 KB | Satisfied |
| (11) | Marking correctness | 0.109 s/0 s/0.413 s | 39,900 KB/105,920 KB | Satisfied |
| (12) | Mark integrity | 0.031 s/0 s/0.425 s | 40,136 KB/106,596 KB | Satisfied |

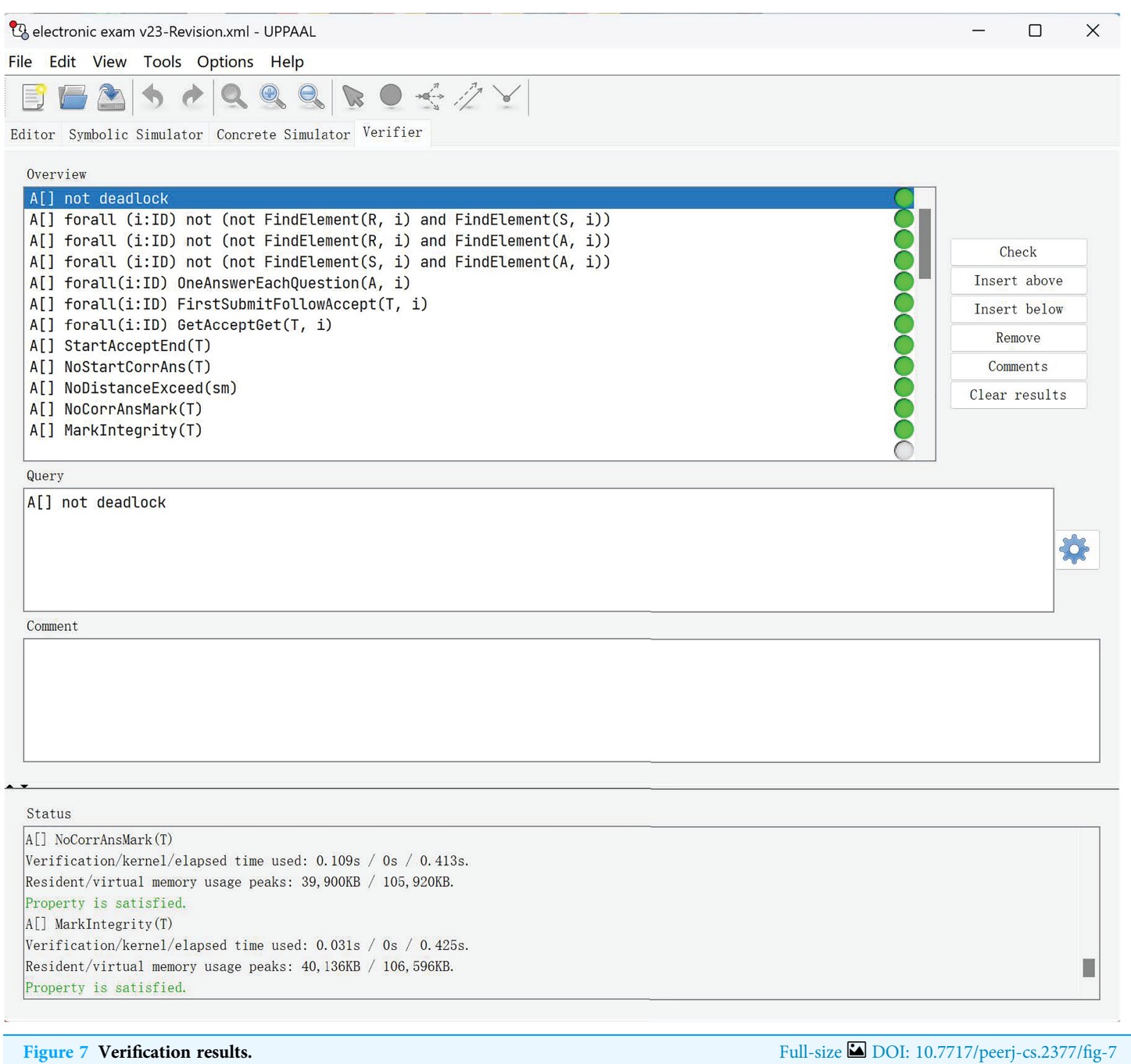

**Figure 7 Verification results.**

ensuring that each candidate can submit only one answer per question (Answer singularity); verifying that all submitted answers are accepted when no examination time expiration (Acceptance assurance); ensuring that candidates receive and answer questions in the correct order (Questions ordering); confirming that the exam is available during the specified period (Exam availability); ensuring that correct answers cannot be modified after the exam starts (Answer-score integrity); detecting any potential cheating by

comparing answers between candidates (Cheater detection); ensuring that correct answers cannot be changed after marking has begun (Marking correctness); and verifying that all answers are marked and candidates are notified of their scores (Mark integrity).

These verification results demonstrate that our model meets the essential requirements for a reliable electronic examination system. The detailed metrics provide insights into the efficiency and resource usage of the verification process.

## Security aspects

The security of electronic examination systems is paramount to ensure the integrity and fairness of the examination process. Our model incorporates several security properties to address potential threats and vulnerabilities:

- **Candidate authentication and eligibility:** Ensures that only registered candidates can participate in the examination and have their answers accepted. This is verified through the Candidate Registration (Property 2) and Candidate Eligibility (Property 3) properties.
- **Data integrity and confidentiality:** Ensures that examination data, such as questions and answers, are protected from unauthorized modification and are only accessible during the examination period. This is verified by the Answer Authentication (Property 4), Answer-Score Integrity (Property 9), and Exam Availability (Property 8) properties.
- **Cheating prevention:** Measures are implemented to detect and prevent cheating. The Cheater Detection property checks for answer similarity between candidates to identify potential cheating (Property 10).
- **Non-repudiation and accountability:** Ensures that actions taken during the examination process can be traced back to specific participants. This is supported by the Acceptance Assurance (Property 6), Marking Correctness (Property 11), and Mark Integrity (Property 12) properties.
- **Order and consistency:** Ensures that candidates proceed in a specified order and that each question receives only one answer. This is verified through the Question Ordering (Property 7) and Answer Singularity (Property 5) properties.

By integrating these security properties into the UPPAAL model, we enhance the overall reliability of the electronic examination system, providing a robust framework for secure digital assessments.

## RELATED WORK

Enhancing the reliability and security of examinations is a crucial prerequisite to ensure accurate assessments of learners' knowledge and abilities. In the realm of electronic examinations, these properties become even more pivotal due to the incorporation of digital techniques. This overview summarizes existing research on the design of electronic examination process and the application of formal methods for electronic examination.

## Design of electronic examination process

The design of electronic examination process encompasses the analysis of learner requirements and the construction of a robust system structure. Many researchers are dedicated to the meticulous process of requirement analysis and the implementation of frameworks for electronic examination systems.

*Muzaffar et al. (2021)* presented a systematic literature review of online examination, identified five leading features, discussed 16 important techniques/algorithms, 11 datasets as well as 21 online exam tools and investigated the participation of countries in online exam research. *Butler-Henderson & Crawford (2020)* reviewed the contemporary literature on online examinations and explored nine key themes, including student perceptions, student performance, anxiety, cheating, staff perceptions, authentication and security, interface, design and technology issues. *Ilgaz & Afacan Adanır (2020)* analyzed learners' academic achievement and perceptions in online exams at a public state university, showing that learners report positive attitudes towards online exams and that there was no statistically significant difference in the students' academic achievement in online and traditional exams. *Jiang et al. (2019)* proposed a web-based online examination system using PHP, Ajax and other technologies, which has been applied to a course involving more than 1,000 students per semester at Guangzhou University of Foreign Studies.

Specifically, certain research endeavors concentrate on the various facets of cheating within electronic examination systems. *Li et al. (2021)* developed an optimization-based anti-collusion approach for distanced online testing (DOT) by minimizing the collusion gain, which can be coupled with other techniques for cheating prevention. *Ngqondi, Maoneke & Mauwa (2021)* used a literature review to understand academic fraud and respective security measures and propose a framework of online examinations for South African universities. *Noorbehbahani, Mohammadi & Aminazadeh (2022)* presented a systematic review of research on cheating in online exams from 2010 to 2021, showed the categorization of the research and discussed topic trends in the field of online exam cheating.

Analyzing requirements and implementing systems are crucial elements; however, the absence of essential quality assurance presents challenges for electronic examination systems that prioritize fairness. The application of formal methods proves beneficial in ensuring the reliability and security of the software process. The integration of formal methods into electronic examination systems significantly contributes to enhancing the overall system.

## Formal methods for electronic examination

Formal methods cover the use of mathematically precise notations to specify and to reason about systems (*Marmsoler, 2022*). These methods, such as automata and Petri nets, prove valuable in enhancing the reliability and security of electronic examinations.

*Kassem, Falcone & Lafourcade (2017)* proposed an event-based model of e-exams, defined several security properties and validated these properties by analyzing real e-exams at UGA using ProVerif and MarQ. However, the explicit interactions between roles in electronic exams are not taken into consideration. In their models, two methods are

employed, namely ProVerif and QEA (Quantified Event Automata). We consistently utilize the UPPAAL automaton to model all these processes.

*Bella et al. (2017)* proposed a secure exam protocol with the design principle of minimizing the reliance on the trusted parties, meeting a series of security requirements and resisting threats. However, this work primarily emphasizes cryptographic aspects, overlooking the absence of a visual and easily understandable modeling approach. A comprehensible model would significantly contribute to the explainability of electronic examinations, benefitting both system designers and teacher/student users.

*Xu et al. (2009)* proposed an approach for modeling online score system using hierarchical colored Petri nets and analyzed concurrency, conflict and causal dependency in CPN Tools. However, this work solely delves into an online score phase and its associated few properties. Our model comprehensively captures the entire processes of electronic examination and verifies a more extensive set of properties across all phases using UPPAAL.

In contrast to their work, our model is both executable and comprehensible. Leveraging the capabilities of UPPAAL, we present general templates and verify critical properties, thereby enhancing the trustworthiness of the electronic examination system.

Table 3 provides a comparative summary of studies in this area, highlighting their focus areas/ contributions, techniques/methods, and limitations. This table illustrates the diverse methodologies appeared in previous studies and demonstrates the novelty of our approach. Our comprehensive electronic examination model uses UPPAAL automata for formal verification, addressing multiple properties such as authentication, cheating prevention, data integrity, and more. Additionally, while our model provides a robust framework for secure digital assessments, we acknowledge the need for it to be more closely aligned with practical cases and the steep learning curve for non-experts.

## DISCUSSIONS

While our study presents a detailed model for electronic examinations using UPPAAL, there are several broader aspects that could benefit from further exploration. These include our experience with UPPAAL, the limitations and potential biases, and the ethical and privacy of the proposed solution.

### Experience with UPPAAL

Our experience with UPPAAL has been largely positive. The graphical interface simplified the modeling process, making it easier to visualize and construct the timed automata. However, the process required adequate effort in overcoming the learning curve, model validation, debugging process, and development time.

- **Learning curve:** Initially, our team spent adequate time familiarizing ourselves with UPPAAL's syntax and semantics, leveraging available tutorials, documentation, and web resources. The C-like scripting language, although powerful, is primarily designed for computer programming and only supports a subset of C language syntax, which limited

**Table 3 Comparison of related work.**

| Reference | Focus area/Contributions | Techniques/Methods | Limitations/Remarks |
|---|---|---|---|
| *Xu et al. (2009)* | Modeling online score system using hierarchical colored Petri nets | Hierarchical colored Petri nets, CPN Tools | Focuses on online scoring phase, does not cover entire examination process or comprehensive properties |
| *Bella et al. (2017)* | Secure exam protocol minimizing reliance on trusted parties | Cryptographic protocols | Primarily cryptographic focus, lacks a visual and easily understandable modeling approach |
| *Kassem, Falcone & Lafourcade (2017)* | Event-based model of e-exams with security properties validated using ProVerif and MarQ | ProVerif, Quantified Event Automata | Lacks explicit interactions between roles, focuses on specific properties |
| *Jiang et al. (2019)* | Web-based online examination system using PHP and Ajax | System implementation using PHP, Ajax | Specific to one implementation, lacks generalizability and comprehensive security analysis |
| *Butler-Henderson & Crawford (2020)* | Review of online examinations covering various themes like student perceptions, performance, anxiety, *etc.* | Literature review | General overview, does not delve into security properties or detailed modeling |
| *Ilgaz & Afacan Adanır (2020)* | Analysis of learners' academic achievement and perceptions in online exams | Empirical study | Focus on academic achievement and perceptions, lacks security aspect |
| *Muzaffar et al. (2021)* | Systematic review of online examination features, techniques, datasets, tools, and research participation | Literature review | General overview, lacks specific focus on security and detailed modeling |
| *Li et al. (2021)* | Optimization-based anti-collusion approach for distanced online testing | Optimization algorithms | Focus on anti-collusion, does not address other security properties comprehensively |
| *Ngqondi, Maoneke & Mauwa (2021)* | Framework for online examinations addressing academic fraud and security measures | Framework proposal | High-level framework, lacks detailed implementation and validation |
| *Noorbehbahani, Mohammadi & Aminazadeh (2022)* | Systematic review of cheating in online exams from 2010 to 2021 | Literature review | Focus on cheating, lacks comprehensive security measures and modeling |
| Our work | Comprehensive electronic examination model ensuring security properties using UPPAAL | UPPAAL, formal verification | Model needs to be more closely aligned with practical cases, steep learning curve for non-experts |

its ease of use and learning. The prior experience of some team members in programming languages and formal methods was helpful in mitigating these challenges.

- **Model validation:** Validating the model required a thorough understanding of both the examination protocol and UPPAAL's features. The tool's robust validation capabilities, including the need to understand Computation Tree Logic (CTL) and timing mechanisms like clocks, greatly assisted in ensuring the correctness and reliability of our model.

- **Debugging process:** Debugging is essential, with UPPAAL's simulation capabilities and clear error messages being very helpful in identifying and correcting issues. The debugging process requires much time and attention to detail, highlighting the importance of understanding UPPAAL-specific primitives such as synchronization.

- **Development time:** The development of the models extended over approximately 5 months, although the actual development time was around 1 month due to other

teaching tasks and research projects. This period included initial learning, model development, extensive testing, validation, and collaborative discussions based on feedback from team members to enhance the correctness and reliability of the models.

In summary, while the learning and implementation phases were intensive, the tools provided by UPPAAL prove invaluable for ensuring the reliability and robustness of our electronic examination system.

## Limitations and Potential Biases

While our study presents a robust model for ensuring reliability in electronic examinations using UPPAAL, it is not without limitations and potential biases that should be acknowledged to provide a balanced perspective.

- **Scope of the model:** Our model focuses on the technical aspects of electronic examinations, specifically the roles of candidates, administrators, invigilators, and examiners. However, it does not account for all possible scenarios, such as those involving multiple administrators or more complex invigilation processes.
- **Simulation constraints:** The simulations conducted were based on a specific set of parameters (*e.g.*, number of candidates, questions, and examination time). These parameters may not cover the full spectrum of real-world examination settings. Therefore, the generalizability of our findings to all electronic examination contexts may be limited.
- **Evaluation metrics:** The reliability of our model was assessed using a set of predefined properties. While these properties are somehow comprehensive, they may not encompass all potential reliability concerns in electronic examinations. Additional evaluation metrics and real-world testing are necessary to fully validate the model.
- **Security questions.** The current model assumes that all participants, including candidates, follow the expected protocols strictly. However, in security scenarios, this assumption may not hold. It is necessary to consider the behavior of candidates who might attempt to exploit the system. Therefore, our future work will include stress-testing the model under conditions where candidates can issue arbitrary requests. This will help us analyze if a candidate can achieve a good grade without submitting correct answers, thereby ensuring the robustness of the examination protocol against such malicious behaviors.

By acknowledging these limitations and potential biases, we aim to provide a more balanced and critical view of our study, paving the way for future research to address these gaps and enhance the reliability and applicability of electronic examination systems.

## Ethical and privacy aspects

In this article, we focus on constructing and verifying a formal model for electronic examinations using UPPAAL, using simulated data for verification. Recognizing the importance of ethical and privacy considerations, we discuss these concerns as follows:

- **Data privacy and consent:** Data collection will be limited to essential information for the examination process. We will implement secure storage methods, including encryption and strict access controls, ensuring only authorized personnel have access. Transparency will be maintained by informing candidates about data collection processes and purposes, obtaining informed consent before participation.
- **Anonymity, confidentiality, and ethical considerations:** Measures will be taken to ensure candidates' personal data remains confidential and anonymous. We will identify and mitigate biases to ensure fairness, maintain transparency about system operations, and reference relevant data protection regulations such as GDPR (General Data Protection Regulation).

As part of our future work, we plan to integrate these ethical and privacy considerations more deeply, incorporating real-world data and addressing the additional challenges this introduces to ensure our model's responsible and ethical application in practical scenarios.

## CONCLUSIONS

This article introduces a UPPAAL-based model for electronic examinations, focusing on both model specification and property specification. The model specification encompasses candidate, administrator, invigilator, and examiner templates. Property specification outlines 12 properties related to electronic examinations, covering aspects like candidate registration and exam availability. Utilizing UPPAAL, all properties are rigorously verified, and the results indicate that our model is reasonably reliable. This offers valuable guidance for system designers and teacher/student users alike.

In future work, we plan to integrate UPPAAL statistical model checking to conduct a more detailed and various evaluation of the developed model, including different scenarios with various numbers of candidates, questions, and scores. We will also integrate multiple administrators and invigilators to better reflect real-world examination scenarios. Additionally, we aim to broaden the scope of examination events to consider non-deterministic candidate automata, potential misbehaviors, and malicious attacks, addressing more complex scenarios. This will include developing non-deterministic automata to interact with the system in arbitrary ways, stress-testing the model by allowing candidates to issue arbitrary requests, and ensuring the examination protocol is robust against denial attacks and misbehaviors. We will also relax current assumptions to develop a more flexible model, investigate the scalability of verification for various parameter sizes, and focus on extensive performance analysis using monitoring methods.

### Funding

This work was supported by the Science and Technology Development Plan Project of Jilin Province of China under Grant No. YDZJ202201ZYTS423, the Fundamental Research Funds for the Central Universities under Grant Nos. 2412022QD040, 2412022ZD018, the Research Fund of Guangxi Key Lab of Multi-source Information Mining & Security under

Grant No. MIMS23-06 and CCF-Huawei Populus Grove Fund under Grant No. CCF-HuaweiLK2023001. The funders had no role in study design, data collection and analysis, decision to publish, or preparation of the manuscript.

### Grant Disclosures

The following grant information was disclosed by the authors:

Science and Technology Development Plan Project of Jilin Province of China: YDZJ202201ZYTS423.

Fundamental Research Funds for the Central Universities: 2412022QD040, 2412022ZD018.

Research Fund of Guangxi Key Lab of Multi-source Information Mining & Security: MIMS23-06.

CCF-Huawei Populus Grove Fund: CCF-HuaweiLK2023001.

### Competing Interests

The authors declare that they have no competing interests.

### Author Contributions

- Wenbo Zhou conceived and designed the experiments, performed the experiments, analyzed the data, performed the computation work, prepared figures and/or tables, authored or reviewed drafts of the article, and approved the final draft.
- Yujiao Zhao performed the experiments, prepared figures and/or tables, and approved the final draft.
- Ye Zhang analyzed the data, prepared figures and/or tables, and approved the final draft.
- Liwen Mu analyzed the data, prepared figures and/or tables, and approved the final draft.
- Yiyuan Wang conceived and designed the experiments, authored or reviewed drafts of the article, and approved the final draft.
- Minghao Yin conceived and designed the experiments, authored or reviewed drafts of the article, and approved the final draft.

### Data Availability

The model is available at GitHub and Zenodo:

- https://github.com/TURTING-BO/An-Electronic-Examination-Model-Based-on-UPPAAL

- Zhou, W. (2024). An Electronic Examination Model Based on UPPAAL. Zenodo. https://doi.org/10.5281/zenodo.12787513.

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
