# Peer review of "Ensuring reliability in electronic examinations through UPPAAL-based trustworthy design"

_PeerJ Computer Science, doi:10.7717/peerj-cs.2377_

## Round 0.1 · original submission · Major Revisions

Dear authors,

Please consider the reviewers' recommendations and comments in order to improve your article. We look forward to your improved version of the article, according to the reviewers' comments.

Best regards.

Reviewer 1 ·

Basic reporting

Very clear, easy to read English. I have only minor suggestions to make some expressions more precise and clearer formatting.

Introduction covers background with sufficient references. I have only a minor suggestion to cite a proper publication rather than a web site.

Good scientific structure (IMRaD). The layout could be improved by moving figures and tables closer to their respective descriptions rather than postponing them to the very end.

The definitions of syntax and semantics are quite informal, but precise and succinct. Since those definitions are not part of the contribution, I feel that they are at the apropriate level for a use case modeling paper such as this one.

The paper provides a fair presentation of the model and also refer to the downloadable model at the Github repository. The model repository at Github also shows a graphical overview of the model with even more queries. I suggest to publish its snapshot at Zenodo (or similar service providing archived DOI references) and then refer to it instead (and include a link to the original repository for future updates).

Experimental design

The paper aims to improve the procedures and protocols of electronic examinations.
The authors choose model-checking as a method to specify and verify the protocol against a series of properties.
UPPAAL has been developed with real-time, embedded and distributed systems and computer communication protocols in mind so it may look unusual in social domains such as this, but model-checking business models and protocols is not neither new nor controversial. Model-checking examination protocol specifically is new to me.
Therefore the paper fits well within model-checking and Computer Science discipline in general.

The paper focuses on reliability of electronic examination protocol, provides a detailed behavior models and appropriate properties. Reliability is a functional property which is difficult to formalise and assess formally, so the formal toolbox is very relevant for this purpose.

There are a few places (notably queries) where I think a reader would benefit with more details on how the used functions are implemented, because those functions seem to refer to a model state in a non-trivial way (there seems to be an implication).

The refered model repository contains all the details need to replicate the results and even more queries.
I have checked the results and it works as described.
The queries could be improved with more details in the comments about the expected outcome and the interpretation ofthe outcome.

Validity of the findings

The reliability and safety of the electronic examination protocol is well argued and shown using formal model-checking techniques.
The model is shown to be sound and the results are easilty replicated using the specified tools.

The conclusions are short and to the point addressing the original problem.
I think an interested reader could benefit from an additional discussion about authors' experience with UPPAAL, perhaps its C-like scripting language (designed for systems programming and not for social engineering). How difficult was it to learn? Validate the model? Debug? How long did it take to make such models?

The conclusions also mention trustworthyness, which I am not so sure, perhaps it's out of scope for this paper.
The issue is that the method assumes that the actors behave just like the automata specify -- that is good enough to show that the examination protocol is viable and free from "natural" faults, but for trust and security requires checking the protocol against active malice, so some of the assumptions about actors need to be relaxed.
For example, a student may have a motive to cheat and acquire good grades without supplying correct answers. For such purpose it would be interesting to construct a completely non-deterministic student automaton which is allowed to interact with the system in arbitrary ways and then check whether "a good grade is reachable", examination protocol is robust against denial attacks and no deadlocks may occur even when student is misbehaving.

Additional comments

I have annotated the paper PDF with specific comments, please see the attachment.

Annotated reviews are not available for download in order to protect the identity of reviewers who chose to remain anonymous.

·

Basic reporting

The paper is not well written and the presentation should be improved in several ways. In addition, there are a few open questions that authors should be addressed to improve the work. Please observe the detailed review of each section attached in the pdf file.

Experimental design

The experimental design is not well justified. The authors use formal analysis using the UPPAAL tool, but there is no comprehensive analysis of the verification properties used in the paper.

Validity of the findings

Comparison with existing works is not detailed and therefore it is not possible to assess the validity of the findings.

Additional comments

The overall presentation of the paper is poor and several important questions have not been addressed.

For instance:

1) Is the simplification considered in the paper relevant so it can be generalized for the problem at hand?
2) There is no information regarding the data and functions used in the paper.
3) Comparison with existing works is shallow.

·

Basic reporting

The article is very interesting and presents a model for electronic examination using timed automata. The English language throughout the article is very good and no major revisions are needed. The list of references is sufficient and corresponds to the text and the presented case studies and the research background. In general the manuscript is well prepared and the content is presented in the best possible way.

Experimental design

The research questions are well defined and are relevant and meaningful. The novelty of study is the use of a real-time system model checker, which is based on timed automata (UPPALL), to model and evaluate the electronic examination processes.

Nevertheless, the conducted simulation experiments are limited and are based on a situation with 2 candidates, 3 questions and 2 scores. A more detailed and highly varied evaluation of the developed model is needed to conclude its efficiency and usability. The article will only gain from the development and evaluation of different scenarios with various numbers of candidates, questions and scores.

Validity of the findings

The conclusions of the experiment are well defined and presented. directions for further work are also presented.

The limited evaluation using just one scenario is significantly reducing the availability of results and the possibility to properly analyze the efficiency and usability of the model.

Nevertheless, the study is presenting the use of a real-time system model checker, which is based on timed automata (UPPALL), to model and evaluate the electronic examination processes, which on its own is a novelty and is worth presenting and discussing.

Additional comments

In general - the article is well structured and presented. The study is interesting and presents the use of a real-time system model checker, which is based on the UPPALL simulation product, to model and evaluate the electronic examination processes. The paper can be improved by the introduction of various simulation scenarios with different number of candidates, questions, and scores.

Reviewer 4 ·

Basic reporting

-The paper introduces an innovative solution by integrating timed automata and the UPPAAL tool to ensure the reliability and security of electronic examinations, presenting a structured and verifiable approach to digital assessment processes. The study addresses a crucial issue in digital education, offering a potentially transformative solution for the trustworthy administration of online exams.
- The introduction provides a good context, but it needs a more detailed discussion on the limitations of current electronic examination systems and the novel contributions of this study.
- The authors are suggested to enhance the grammar and sentence structure throughout the article to improve readability and comprehension.
- Ensure consistency in the usage of abbreviations and their full forms throughout the article.
- The quality of the figures need to be improved, especially Fig. 3, 4, and 6 are not readable. Authors are suggested to consider enhancing the quality of the figures to ensure they are more visually effective and informative.
- A subsection can be added to discuss the security aspects of the proposed solution.
- While the paper references several related works, it could benefit from a more in-depth and critical review of existing literature. To this end, a table can be added to compare related works and the proposed solution better to highlight the contributions and novelty of the work.
- A discussion of the study's limitations and potential biases can be added. Addressing these aspects would provide a more balanced perspective.
-On page 2, line 79, the authors provided the link to the code inside the text. It needs to be given in reference format.

Experimental design

- The methodology is rigorous and employs UPPAAL effectively for modelling and verification. However, the explanation of some technical details needs to be more accessible to readers less familiar with UPPAAL.
- The research question is clear, but its significance and impact should be emphasized more.
- While methods are described in detail, the inclusion of more examples or case studies would further aid in replication.
- A section on the validation and testing procedures, specifying how the model performance was evaluated, can be added.

Validity of the findings

- A discussion on the ethical and privacy aspects of the proposed solution can be added.
- A more extensive discussion on the evaluation metrics and their limitations can be added to provide a clearer understanding of the proposed solution’s robustness.
- Table 2 provided the verification results of the work but the results section of this table need to be explain in detail, especially what is the satisfied criteria. The current version does not explain it in detail.
- “Validation and verification in UPPAAL” section needs to be explained in more detail.

---

## Round 0.2 · Minor Revisions

Dear authors,

One of the reviewers indicates that some minor changes need to be made to make your article suitable for publication in PeerJ Computer Science. Resubmit the article after making those minor changes.

We look forward to your article with those changes.

Best regards,

Reviewer 1 ·

Basic reporting

I see that my previous comments have been fixed, so I will not reevaluate the reporting and findings.
The overall paper quality has improved a lot, especially the related work and presentation.

Experimental design

See my original assessment.

Validity of the findings

See my orignal assessment.

Additional comments

Some of my comments were taken a bit too literally, in particular lines 292-296 are superfluous since the authors do not use this math notation (my humble suggestion was either to use math notation or to use UPPAAL notation, authors chose to stick to UPPAAL notation, which is a fine alternative).
I would still recommend to use a proper monospace font for technical text so that <> and [] sequences would look closer to diamond and box. The code listings do use a proper monospace font, so that's nice.
Sentence on line 302 could be the first sentence in "Property specification" paragraph as it introduces TCTL. TCTL was created by Rajeev Alur in his thesis https://dl.acm.org/doi/10.5555/143902
"cend[i]" on Line 378 needs monospace font.
Code on line 919 looks unused/dead, could be removed.

Reviewer 4 ·

Basic reporting

The authors successfully addressed all my previous comments. I do not have further comments.

Experimental design

The authors successfully addressed all my previous comments. I do not have further comments.

Validity of the findings

The authors successfully addressed all my previous comments. I do not have further comments.

---

## Round 0.3 · Minor Revisions

Dear Authors,

Please accept our apologies for asking for more changes. They are necessary for the article to be published. These minor changes are as follows:

- Refer to the sections of the paper by their name, not by their number, as when defining their structure in lines 114-119.

- On line 199, move the title “Editor” to the next page.

We look forward to your article with those changes.

Best regards,

Emilia

---

## Round 0.4 · accepted · Accept

Dear Authors,

I am pleased to inform you that your article has been accepted for publication in PeerJ Computer Science.

Best regards,

Emilia Cambronero
Academic Editor for PeerJ Computer Science